# Effect of Three Diets (Low-FODMAP, Gluten-free and Balanced) on Irritable Bowel Syndrome Symptoms and Health-Related Quality of Life

**DOI:** 10.3390/nu11071566

**Published:** 2019-07-11

**Authors:** Danilo Paduano, Arianna Cingolani, Elisabetta Tanda, Paolo Usai

**Affiliations:** 1Gastroenterology, Department of Medical Sciences, Policlinico Universitario di Monserrato, University of Cagliari, 09042 Cagliari, Italy; 2Thoracic Vascular Surgery, Department of Surgical Sciences, Policlinico Universitario di Monserrato, University of Cagliari, 09042 Cagliari, Italy

**Keywords:** IBS, FODMAPs, diet, diet treatment, irritable bowel syndrome, gluten-free diet, balanced diet, quality of life

## Abstract

Several studies have reported some efficacy of diets low in fermentable carbohydrates (Fermentable Oligo-, Di-, Monosaccharides and Polyols (FODMAPs)) in Irritable Bowel Syndrome (IBS). There is no evidence of its superiority compared to gluten-free and balanced diets in improving IBS patients’ quality of life (QoL). The aim of this study is to assess whether different diets can improve QoL in IBS. Forty-two patients with IBS, according to Rome IV criteria, were enrolled. Low-FODMAP, gluten-free and balanced diets were proposed to each patient in the same succession. Each diet was followed for 4 weeks. The Bristol Stool Scale, the Visual Analogue Scale (VAS) for bloating and abdominal pain, and the SF12 questionnaire for health-related quality of life were applied at the beginning and at the end of each diet. Twenty-eight of the forty-two patients completed all the three diets. All the three diets reduced symptom severity (*p* < 0.01), bloating (*p* < 0.01) and abdominal pain (*p* < 0.01), and improved quality of life (*p <* 0.05); 3% of patients expressed a preference for the low-FODMAP diet, 11% for the gluten-free and 86% for the balanced diet (*p <* 0.01). The balanced diet improves QoL and VAS pain, provides an adequate quantity of FODMAPs and is more appreciated by patients. For these reasons, the balanced diet could be recommended to patients with irritable bowel syndrome.

## 1. Introduction

Irritable bowel syndrome is a functional disorder characterized by the presence of recurrent abdominal pain at least once a week for the last three months, associated with one or more of the following: (1) correlation between symptoms and evacuations; (2) variation in the frequency of evacuations; (3) variation in the solidity of stools. The onset of symptoms must precede the diagnosis by at least 6 months [1]. Even if IBS is a benign condition, it leads to chronic relapsing symptoms, with fatigue, depression, and anxiety. IBS is estimated to affect 11.2% of the world’s population, most frequently women [2], mainly people between 20 and 30 years old, even if recent studies show a remarkable prevalence even in older populations [3]. This prevalence is very similar across different countries, despite significant differences in lifestyle [4]. The IBS pathogenesis is mainly unknown, even though some hypotheses have been posited regarding visceral motility hyper-activation, visceral hypersensitivity, alterations in the central nervous system, as well as serotoninergic pathways, psychological illness and stress and mucosal inflammation [5]. Subjects with IBS have a lower quality of life, are more likely to use health service resources, and show a higher work absenteeism than healthy controls [6]. The disease can be clinically managed through nutritional interventions and appropriate changes in lifestyle, and through pharmacologic symptom-targeted or psychological therapies [3,4]. Some pharmacological treatments have been proposed to alleviate IBS-related symptoms, although they could be limited by side effects and they are not to be optionally preferred for a long-term treatment. Nowadays, pharmacological therapies are mostly symptom-targeted and they have not shown a consistent effectiveness [7]. As 84% of patients with IBS believe that some foods are responsible for triggering the abdominal symptoms, some dietetic interventions have been proposed. Foods containing gluten seem to cause symptoms in one out of four patients [8]. Dietary issues represent only a part of the pathophysiological network of IBS. However, dietary counselling is a major component of the therapeutic management in these patients [9].

The effect of gluten restriction in IBS is unclear. Evidence from uncontrolled studies [10,11] and a controlled trial [12] suggest that a gluten-free diet leads to symptomatic benefits in IBS with diarrhea (IBS-D) patients with HLA-DQ2 or HLA-DQ-8 genotype. Nevertheless, a short-term, double-blind, placebo-controlled, cross-over trial on background diets failed to show any benefit, even if anticipatory nocebo effects are to be considered [13]. Further clarification of the role of gluten restriction in managing IBS symptoms is required [14]. Growing evidence shows how other substances such as FODMAPs (Fermentable Oligo-, Di-, Monosaccharides And Polyols) can contribute to generate symptoms in IBS patients [15] (abdominal pain or discomfort, bloating, nausea, heartburn, drowsiness [13,16]); however, it is uncertain whether they really cause this disease [17]. High costs and the absence of etiologic therapies are leading researchers to focus more attention on nutritional therapies and, in recent years, several studies have reported the effectiveness of a low-FODMAP diet in the treatment of IBS patients [17,18,19,20,21]. FODMAPs are short-chain carbohydrates, lowly absorbed in the small bowel and highly fermentable by gut flora. In detail, they consist of fructose monosaccharide, lactose disaccharide, the oligosaccharides fructan and galactan, and polyols like sorbitol, mannitol, xylitol and maltitol [21]. Some data suggest that a gluten-free diet reduces diarrhea in IBS-D patients, and that patients affected by abdominal pain and bloating reported a resolution of symptoms after 6 weeks of a gluten-free diet [22]. Since many cereals containing gluten also contain fructans (FODMAPs), some authors claim that it is their simultaneous reduction with the gluten-free diet that determines the improvement of abdominal symptoms. Therefore, gluten could induce symptoms only in the presence of a high FODMAP content. The aim of this study is to compare the efficacy of a low-FODMAP diet, a gluten-free diet and a balanced Mediterranean diet in terms of improving IBS patients’ health-related quality of life, increasing stool solidity, reducing abdominal bloating and pain, and assessing the feasibility of the diets through patients’ acceptance and adherence to each diet.

## 2. Materials and Methods

### 2.1. Patients

Between September and October 2018, 42 consecutive outpatients aged between 18 and 45 years (7 men, 35 women, with a mean age of 28.62 ± 6.86) of the Internal Medicine Department at the University of Cagliari—Gastroenterology Clinic—were enrolled according to Rome IV criteria. Patients were categorized according to the frequency and type of stools: IBS-D, IBS with diarrhea (22pt), IBS-C, IBS with constipation (11pt), IBS-M, mixed form (5pt), IBS-U, undefined (4pt). According to the analysis of their previous food habits through 24 hr recall, 26% of patients did not normally eat breakfast, 38% of patients did not normally eat snacks and all of them normally accumulated the main part of their daily calories at their evening meal. Exclusion criteria were: gastrointestinal organic diseases, other functional diseases, clinically significant systemic diseases, established food allergies, eating disorders and major abdominal surgery. During this study, enrolled patients were not allowed to take laxative, prokinetics, antispasmodics, antidiarrheal drugs or antibiotics. The study was conducted in agreement with the Helsinki Declaration. Diets were administered by a dietician according to the following order: low-FODMAP (reduction of all FODMAPs with gluten allowed), strict gluten-free (GFD) and balanced mediterranean diet. The low-FODMAP diet consisted in reducing all FODMAPs whilst maintaining gluten-containing foods. The Balanced Mediterranean diet maintained FODMAPs and gluten-containing foods. Each diet was structured as follows: three main meals (breakfast, lunch, dinner) and two light snacks between them. In order to reduce the carryover effect of a diet, it was considered appropriate to suspend a diet for a month and then start the next one. Each diet was followed for 4 weeks. At enrolment time and after each diet, the same questionnaires were administered. The Bristol stool scale [23] to evaluate stool features and the Visual Analogue scale [24] for abdominal bloating and recurrent abdominal pain of <h duration were used. The IBS Severity Scoring System [25] to evaluate the disease severity, and IBS quality of life (QoL) [26,27] and SF12 for the quality of life [27,28] were performed. At the end of the study, only patients who had completed all three diets were asked to answer a questionnaire about acceptance levels, independently of the results. At enrolment time, a recent blood count, CRP, ESR, tTG IgA and IgG, total IgA, TSH, FT3, FT4, fecal calprotectin and an H2 breath test for lactose were examined. Anthropometric data were obtained with an OMRON BF511 bioelectric impedance analyzer, TRAILITE TL-LSC100 dynamometer and BMI measuring tape; the weekly physical activity and food habits were examined through a 24 h recall method. Patients were informed that the study aimed to evaluate the efficacy of three feasible diets for IBS, that each of them had to be followed for 4 weeks, and that at the end of every diet the questionnaires would have to be performed again. It was also explained to the patients that the first diet would reduce fermentable substances, the second would withdraw gluten, and the third would balance nutrients over a 24 h period. All patients always encountered the same professionals and the word “FODMAP” was never used. Data were analyzed through the “Progetto dieta” software, to obtain: the daily calories burned in kcal, the percentage and weight of daily proteins, carbohydrates, and lipids, and the alcohol intake. The number of daily fruit and vegetable portions was highlighted, considering 5 portions a day as the ideal. The study population was found to have an average basal metabolism of 1304 ± 179 kcal and a daily consumption of 1969 ± 634 kcal (Table 1). These were matched with the anthropometric data (Table 2) to evaluate the daily calories needed and to point out the incorrect alimentary habits, in order to provide adequate and personalized dietary regimens. This study was approved by the Independent Ethics Committee of Policlinico Universitario di Cagliari, Duilio Casula (PPG/2018/16315). Written informed consent was obtained from all of the participants upon explanation of the study purpose.

### 2.2. Diets

The assigned dietary regimens were obtained through the “ProgeoNutrigeo” software, according to the 2104 LARNs (Referral Levels of Nutrients and Energy Assumption), periodically revised from SINU (Italian Society of Human Nutrition), specifically oriented to the Italian population [29]. The daily macronutrient intake according to 2014 LARNs is: 45–60% glucides, 20–35% lipids, and 10–15% proteins.

### 2.3. Statistical Analysis

The estimated minimum sample size was 18 subjects. Considering a 15% probability of drop out, it was estimated as necessary to recruit at least 21 patients. Given the number of patients with IBS that refers to our division, we were able to recruit 42 patients. The questionnaire scores were analyzed to determine if the differences between the enrolment time and the endpoint after the three diets were statistically significant; the analysis was performed using the “R” software developed by the Statistics Department of Auckland University. The first step was to verify the normality of data distributions. Given that our population was smaller than 50 units, normality was evaluated with the Shapiro–Wilk test, with a confidence interval of 95%. The results achieved after implementation of the three diets were further compared to the basal ones (control group). Only when both the control group and the comparison group data had normal distributions, and their variances were estimated to be equal through the Fisher test, was the comparison realized using a Student’s t-test for independent samples. In all other cases, we used the Wilcoxon test, which does not need normal distributions of data in order to be applied. At the end of our study, patients were asked to express their preference out of the three diets, independently from the results. The three percentage values were compared with a Chi-Square Test to evaluate the statistical significance. In all the analyzed cases, the p value was 0.05, with a confidence level of 95% for the comparison. For IBS-QoL, Visual Analogue Scale (VAS) bloating, VAS pain and Irritable Bowel Syndrome Severity Scoring System (IBS-SSS), the low-FODMAP diet was compared to the gluten-free and balanced diets in order to evaluate any possible statistical differences between the results.

## 3. Results

Of the 42 enrolled patients, 34 followed the low-FODMAP diet; of these, 30 followed the gluten-free, and 28 followed the balanced diet as well, thus carrying out all the three diets. The dropouts were, respectively, eight for the low-FODMAP diet (thus for the entire study), four for the gluten-free diet and two for the balanced diet (Table 3).

### 3.1. Gastrointestinal Symptoms

#### 3.1.1. Stool Type According to Bristol Stool Chart

Stool solidity, based on the Bristol scale, had a basal mean value of 6. After the low-FODMAP diet, the mean value moved to type 4, which is considered normal. The difference was statistically significant (*p* = 0.03), showing the efficacy of a low-FODMAP intake in normalizing stool solidity. In detail, of the 16 IBS-D patients, 14 showed improvement (that is to say, a movement towards the 4th type on the Bristol scale) and two had no changes. Of the 10 IBS-C patients, seven had an improvement while three had no changes in stool solidity. Of the five IBS-M patients, two showed an improvement, one patient worsened, two patients had no changes, and one of them had Type 4 at basal. Of the three IBS-U, none had changes in the stool solidity, but they all had Type 4 at basal. Overall, 79% of patients (27 of 34) showed a trend to reach Type 4 on the Bristol Stool chart after 4 weeks of the low-FODMAP diet. It is worthwhile to clarify that the low-FODMAP diet showed a limited efficacy in IBS-C in some previous studies [30]. No statistically significant difference in stool solidity was observed after the gluten-free and balanced diets.

#### 3.1.2. Abdominal Bloating

All three diets produced a statistically significant decrease (*p <* 0.01) in VAS bloating versus the basal values. The three diets were compared and the low-FODMAP diet showed superiority in decreasing abdominal bloating, in particular when compared to the gluten-free diet (Table 4).

#### 3.1.3. Abdominal Pain of <24 h Duration

All three diets showed efficacy in reducing the duration of abdominal pain to less than 24 h (*p <* 0.01). Durations of longer than 24 hr abdominal pain were not evaluated because only one patient showed an occasional persistence of the pain during the day (Table 5). The low-FODMAP diet did not show superiority compared to the other two diets in reducing abdominal pain (Table 4).

#### 3.1.4. Disease Severity According to IBS-SSS

The three diets show the same efficacy (*p <* 0.01) in reducing the disease severity, evaluated through the IBS-SSS questionnaire considering the last 10 days before its administration (Table 4; Table 5).

### 3.2. Quality of Life

#### 3.2.1. SF12

As for quality of life, evaluated through SF12, the low-FODMAP and gluten-free diets showed an improvement both on the physical (PCS) and the mental (MCS) side (*p <* 0.05). The balanced diet showed an improvement only for MCS (Table 5).

#### 3.2.2. IBS-QoL

Overall, the QoL variable, evaluated through the IBS-SSS-specific questionnaire, was modified to a statistically significant extent after each of the three diets (*p <* 0.05) (Table 5). Comparing the diets, the low-FODMAP diet did not show greater efficacy in improving the quality of life, compared to the others (Table 4). From the analysis of the eight subclasses of IBS-QoL, the three diets proved to induce a significant improvement in dysphoria, daily activity interference, body image, and worries about health and social relationships, while they did not show any statistically significant difference in food refusal, sexuality and relationships.

### 3.3. Adherence Index

The balanced diet obtained the highest adherence index. The results were 86% for the balanced diet, 11% for the gluten-free diet, and 3% for the low-FODMAP diet; *p <* 0.01 at χ^2^ test.

## 4. Discussion

The reported findings need to be considered in light of some limitations: (1) the three diets were offered to patients in the same sequence, without randomization, no blinding of health care professionals and patients was performed; (2) the results are based on patient’s reports, which could have been influenced by the fact they knew what each diet provided; (3) we could not verify and measure patient’s real compliance to the diets, as they were outpatients; (4) the study population is unbalanced towards the female sex, this could have led to bias, but it is a direct representation of the demographic characteristics of IBS patients that refers to our department. These aspects need to be improved in future research.

Low-FODMAP, gluten-free and balanced diets proved to be effective in reducing abdominal pain and bloating, and in improving the quality of life in IBS patients. The low-FODMAP diet did not show superiority versus the other two dietary regimens in reducing abdominal pain, but it was the only regimen to regularize the bowel functions by reaching the 4th grade of the Bristol Stool Scale, far beyond the improvement we expected from the deprivation of osmotically active food in patients with IBS-D. The increase in stool consistency observed mostly among patients with IBS-D could be justified by the decrease in concentration in the intestinal lumen of osmotically active substances such as short-chain fatty acids. These compounds, in fact, derive from the fermentation of FODMAPs by the colonic bacterial flora. Furthermore, in the low-FODMAP diet, we also observed a significant improvement in VAS pain and VAS bloating in all IBS types. This improvement could be due to a decreased production of gases such as hydrogen and methane by the colonic bacterial flora [14,31,32,33]. In fact, FODMAPs increase microbial fermentation and gas production and, therefore, colonic volume, thus increasing pain and bloating. It is also worth specifying that healthy individuals have the same physiological responses to FODMAPs as IBS sufferers, but in the latter, visceral hypersensitivity may worsen symptoms. [5]

In particular, the low-FODMAP diet leads to a considerably reduced intake of prebiotic fructans and galactooligosaccharides (GOS) [20,34], and, therefore, a sizeable reduction in the substrate available for colonic fermentation. On the other hand, FODMAPs represent an important substrate for the synthesis of short-chain fatty acids such as butyrate. Significant dietary restrictions can alter the composition and functioning of the gastrointestinal microbiota [14]; therefore, excessive dietary restrictions might cause nutritional deficiencies in otherwise ‘healthy’ patients.

After GFD, we observed an improvement in symptoms, in particular, VAS bloating, VAS pain and severity of disease (IBS-SSS), with a smaller improvement in bloating compared to the low-FODMAP diet. Currently, the role of GFD in patients with IBS is unclear, while in some studies it has actually been demonstrated to generate symptoms in individuals with IBS [22]. In a subsequent study, with the same group of individuals as in [22], when GFD was compared to placebo, no effects of gluten were demonstrated in patients with self-reported NCGS and IBS [13]. The improvement in symptoms may be determined not only by the exclusion of gluten but also from the limited use of packaged foods also containing fermentable sugars such as fructans, with both intestinal and extra-intestinal effects. Thus, a combined exposure to gluten and other components of wheat, such as ATIs, lectin, wheat germ agglutinin and fructans, may worsen symptoms through possible synergistic actions [35,36]. In addition, a GFD may in some cases, present a financial burden, and the GFD has been linked with a higher risk of nutritional inadequacies and a higher incidence of coronary heart disease, attributed to lower intakes of wholegrains [37], but it should be noted that risks of nutritional deficiencies and inadequate fiber intake recur in all diet therapies.

The balanced diet obtained the higher acceptance level and proved to be effective in improving QoL and VAS pain. Most of the study population had IBS-D, in which diarrhea and abdominal bloating lead to significant discomfort; therefore, the superiority of a balanced diet in improving QoL may suggest the presence of further important issues. The balanced diet is mostly focused on increased fiber intake, improving food habits in patients who are used to skipping breakfast and snacks, and also, on redistributing meals, calories and FODMAPs over the 24 hr period. The improvement that was also observed among patients who regularly consumed breakfast, morning snack, lunch, afternoon snack and dinner, could derive from the redistribution of FODMAPs over five meals. This group of patients, in fact, preferred FODMAPs to be consumed in one meal, either lunch or dinner, maybe due to a misinterpretation of the Mediterranean diet, concentrating the supply of fermentable sugars in a limited space of time. The balanced diet, by varying and harmonizing the meals, prevents patients from an excessive FODMAP intake. This diet ensured, more so than the other two, the maintenance or the achievement of a healthy body weight. It satisfies the Mediterranean diet criteria, provides an adequate quantity of FODMAPs, which are essential in the production of short-chain fatty acids (SCFT), and it is more appreciated by patients [38].

Short-chain fatty acids are particularly important for colon health as they are the primary energy source for colonic cells and have anti-carcinogenic, as well as anti-inflammatory properties [37], which are important for keeping colon cells healthy [39,40]. Butyrate inhibits the growth and proliferation of tumor cell lines in vitro, induces the differentiation of tumor cells, producing a phenotype similar to that of a normal mature cell [41], and induces apoptosis or programmed cell death of human colorectal cancer cells [42,43]. Butyrate inhibits angiogenesis by inactivating Sp1 transcription factor activity and downregulating VEGF gene expression [44]. Last but not least, the low-FODMAP diet represents a financial burden for patients, and, in our experience, it obtained the lowest acceptance level and did not show differences in improving QoL compared to other diets in IBS patients. Many patients with IBS believe that their symptoms are triggered by specific foods [45,46]. Although this causative relationship is difficult to prove, the majority of patients limit or exclude food items from their diet, and a small but relevant proportion of them have a poor response and an inadequate diet [37,46]. After taking an accurate dietary history, most patients will realize that a given foodstuff, which apparently caused symptoms on one occasion, was well tolerated in many other instances [35]. Therefore, in general, most patients with IBS could (and should) eat a balanced diet without restrictions [9].

The dropout analysis showed that patients not following the first diet that we proposed to them, and therefore, the entire protocol, had, at the enrolment time, a higher QoL, a lower VAS bloating and disease severity, higher instruction and lower disoccupation compared to the mean values of the examined population. The mild clinical features experienced by the patients and their demographic and cultural characteristics could be a cause of their low adherence to the protocol, since the patient may have considered the treatment to be less necessary, or the suggested dietary recommendations to be excessively burdensome, considering their mild symptoms. Patients had symptoms within the 4 weeks interval of diet washout thus a four-week regimen for each diet might be enough to observe the results of the on-going diet, and no longer exhibit the effects of the previous one. An immediate osmotic effect of FODMAPs is described [29], while we still do not have much evidence about the timing of a gluten-free diet in people affected by IBS. Similar studies worldwide have been managed through 3–4-week dietary regimens, and also consider this to be an adequate washout interval [35].

## 5. Conclusions

Considering the results obtained and patients’ dietary habits, it is probably more appropriate to recommend a diet that contains FODMAPs, but adequately distributed in different meals throughout the day, avoiding their overload, and with a correct distribution of calories. This investigation highlighted that patients often avoid breakfast and intermediate meals in the morning and afternoon, and they concentrate most of the caloric needs at dinner (evening meal). A balanced Mediterranean diet, while guaranteeing an adequate daily supply of carbohydrates, proteins, lipids and fibers, prevents patients from excessive FODMAP overload by dividing the total intake into five meals. In this way, it avoids the excessive load of FODMAPs at lunch and dinner, which is often responsible for the exacerbation of symptoms, and it subdivides FODMAP intake, in smaller quantities, into the five daily meals. It also avoids an excessive economic burden, is well accepted by patients and is easier to follow compared to the other two proposed diets, in the short-term period of four weeks. In conclusion, IBS is a heterogeneous entity and the increasing knowledge of its pathophysiology supports the potential of dietary therapies. Our work is evidently based on real life and everyday clinical experience and may be open to bias, but we believe it is appropriate to conduct investigations in this way because currently, the majority of dietary therapies still focus mostly on symptoms and not enough on patients’ quality of life.

## Figures and Tables

**Table 1 nutrients-11-01566-t001:** Dietary habits of the analyzed Irritable Bowel Syndrome (IBS) population.

Proteins, %	16 ± 5
Proteins, g	76 ± 25
Lipids, %	28 ± 8
Lipids, g	61 ± 26
Carbohydrates, %	55 ± 10
Carbohydrates, g	270 ± 110
Alcohol, %	0.06 ± 0.02
Fruit and vegetable portions, number of	2.6 ± 1.2
Basal metabolism, kcal	1304 ± 179
Daily consumption, kcal	1969 ± 634

**Table 2 nutrients-11-01566-t002:** Mean anthropometric data of the analyzed IBS population.

Weight, Kg	57 ± 13
Height, cm	162 ± 8
Body Mass Index, Kg/m^2^	22 ± 4
Fat mass, %	27 ± 10
Lean body mass, %	30 ± 5
Basal metabolism, kcal	1310 ± 180
Visceral fat, %	4 ± 2
Arm circumference, cm	26 ± 4
Waist circumference, cm	79 ± 11
Dynamometer hand strength, Kg	27 ± 9

**Table 3 nutrients-11-01566-t003:** Adherence to the dietary regimen.

	Low-FODMAP	Gluten-Free	Balanced
completed	34	30	28
dropout	8	4	2

**Table 4 nutrients-11-01566-t004:** Clinical and statistical significance of comparisons between the low-FODMAP diet and the other two diets.

	Low-FODMAP vs. Gluten-Free	Low-FODMAP vs. Balanced
VAS bloating	3 ± 2 vs. 4 ± 2 (*p* = 0.04)	3 ± 2 vs. 4 ± 2 (*p* = 0.01)
VAS pain	2 ± 2 vs. 3 ± 2 (*p* = 0.08)	2 ± 2 vs. 3 ± 2 (*p* = 0.17)
Disease severity (IBS-SSS)	16 ± 8 vs. 19 ± 9 (*p* = 0.19)	16 ± 8 vs. 17 ± 7 (*p* = 0.44)
IBS-QoL	83 ± 14 vs. 79 ± 14 (*p* = 0.26)	83 ± 14 vs. 81 ± 11 (*p* = 0.27)

**Table 5 nutrients-11-01566-t005:** Clinical and statistical significance of comparisons with basal values. Score values: Bristol stool chart (1–7); Visual Analogue Scale (VAS) bloating and VAS pain (0–10); Irritable Bowel Syndrome Severity Scoring System (IBS-SSS) (0–50); physical (PCS) and mental (MCS) side (0–100); IBS quality of life (QoL) (0–100); dysphoria, social activity interference, body image, health worries, food refusal, social relationships, sexuality and relationships (0–100).

	Basal	Low-FODMAP	Gluten–Free	Balanced
Bristol stool chart		*p* = 0.03	*p* = 0.13	*p* = 0.31
VAS bloating	6 ± 3	3 ± 2 (*p* < 0.01)	4 ± 2 (*p* < 0.01)	4 ± 2 (*p* < 0.01)
VAS pain	5 ± 2	2 ± 2 (*p* < 0.01)	3 ± 2 (*p* < 0.01)	3 ± 2 (*p* < 0.01)
Severity of disease (IBS-SSS)	29 ± 10	16 ± 8 (*p* < 0.01)	19 ± 9 (*p* < 0.01)	17 ± 7 (*p* < 0.01)
PCS	43.9 ± 7.8	49.9 ± 7 (*p* < 0.01)	47.7 ± 7.9 (*p* = 0.04)	47.4 ± 7.7 (*p* = 0.06)
MCS	38 ± 9.9	46.9 ± 8.4 (*p* < 0.01)	43.9 ± 9.2 (*p* = 0.01)	46.9 ± 10.3 (*p* < 0.01)
IBS-QoL	70 ± 17	83 ± 14 (*p* < 0.01)	79 ± 14 (*p* = 0.01)	81 ± 11 (*p* < 0.01)
Dysphoria	69 ± 22	86 ± 13 (*p* < 0.01)	81 ± 14 (*p* = 0.01)	82 ± 11 (*p* < 0.01)
Social activity interference	67 ± 21	80 ± 19 (*p* < 0.01)	76 ± 20 (*p* = 0.03)	79 ± 17 (*p* = 0.01)
Body image	63 ± 20	77 ± 16 (*p* < 0.01)	75 ± 15 (*p* < 0.01)	78 ± 13 (*p* < 0.01)
Health worries	70 ± 20	82 ± 17 (*p* < 0.01)	82 ± 16 (*p* < 0.01)	82 ± 12 (*p* < 0.01)
Food refusal	50 ± 37	64 ± 29 (*p* = 0.14)	61 ± 28 (*p* = 0.38)	65 ± 24 (*p* = 0.13)
Social relationships	74 ± 21	87 ± 12 (*p* < 0.01)	85 ± 14 (*p* = 0.01)	87 ± 11 (*p* < 0.01)
Sexuality	88 ± 17	93 ± 11 (*p* = 0.23)	92 ± 16 (*p* = 0.17)	96 ± 6 (*p* = 0.11)
Relationships	81 ± 19	88 ± 15 (*p* = 0.67)	87 ± 15 (*p* = 0.79)	86 ± 16 (*p* = 1.00)

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
