# Peer review of "Effect of Three Diets (Low-FODMAP, Gluten-free and Balanced) on Irritable Bowel Syndrome Symptoms and Health-Related Quality of Life"

_nutrients, 2019, doi:10.3390/nu11071566_

Round 1

Reviewer 1 Report

The authors examined symptoms and QoL in 42 IBS patients at enrollment and after 4 weeks each of a low-FODMAP diet, a gluten-free diet and a „balanced mediterranean diet“, in this order, with 4 weeks „free“ diet in between.

The trial appears to be registered only with a local registry, and review by an ethics committee is not stated. Inclusion and exclusion criteria (e.g. age?) and interventions are not well defined. It is unclear, if a consecutive or a convenience sample was studied, and sample size is not justified. Order and carryover effects are not addressed, so the effects measured cannot be meaningfully compared, but for most parameters only comparative statistical significance  and no measurements are reported.

Author Response

Dear  reviewer,

Thank you very much for the constructive feedback you provided regarding our manuscript, “Effect of three diets (Low in FODMAPs, Gluten-free and Balanced) on Irritable Bowel Syndrome symptoms and health-related quality of life.”

As suggested the following articlehas undergone English language editing by MDPI. The text has been checked for correct use of grammar and common technical terms, and edited to a level suitable for reporting research in a scholarly journal. We agree with your suggestions and we tried to clarify as much as possible all the important issues you indicated:

1. the study was reviewed and authorized by an independent ethics committee, we specified it in line 119.

2. we enrolled consecutive patients between the ages of 18-45 (line 80) who met the ROME IV criteria (line 82). Moreover we have better specified the exclusion criteria as suggested (line 87).

3. a consecutive sample was studied (line 80) and a minimum sample size required of 18 subjects was determined, considering the possibility of diet interruption or follow-up loss for 15% of patients, it was necessary recruit at least 21 patients. Given the number of patients with this disorder that belong to the Department of Gastroenterology of the AOU of Cagliari, it has been estimated that it will be possible to recruit about 40 patients.

4. the carryover effect is an important issue and we tried to clarify this aspect in line 95.

5. We added clinical values in the tables 4 and 5 as suggested.

We hereby resubmit our manuscript for a secondary evaluation. Thank you once again for your consideration of our paper. 

Sincerely

Dr. Danilo Paduano

Reviewer 2 Report

In this manuscript, the authors bring a novel comparison of the low FODMAP diet, GFD and ‘balanced’ diet in IBS, which brings interesting new data to the literature. Whilst the authors do highlight a number of interesting points, further detail is required to highlight the new information which is brought by this manuscript. I have outlined some of my comments below: 

1.    Line 49 – Pharmacological therapies are stated to have not shown consistent evidence in IBS – please highlight evidence for this 

2.    Line 52 – The authors state that dietetic therapies are safe and free from collateral effects – there are true concerns of these dietetic therapies, such as the low FODMAP diet and GFD having the potential risk of nutritional deficiencies, as well as some studies highlighting heavy metal accumulation on a GFD. Whilst this is mentioned in the discussion, I am unsure that the statement of dietary therapies being safe can be qualified 

3.    Line 58 – whilst the study mentioned (ref 12) failed to show benefit of a GFD, it maybe worthwhile to discuss limitations of this study. Due to the nature of the design of this study, patients may have had an anticipatory nocebo effect, as they were receiving varying levels of gluten. 

4.    Line 60 – The authors highlight FODMAPs as other gluten free substances. Whilst a low FODMAP diet may reduce fructan content like a GFD, not all gluten free substances are FODMAPs

5.    Line 71 – please check this study – was this 6 weeks of a GFD rather than the 6 months as stated?

6.    Line 81 – consider placing standard deviation with mean when presenting this data

7.    Line 84 – how were previous food habits analysed – please state method 

8.    Line 93 – who delivered these diets – was this delivered by a dietitian? Also, what aspect of the low FODMAP diet was delivered? Was this the strict reduction of all FODMAPs phase?

9.    Line 150 – Please clarify the mean basal value of stool solidity? Was this 5 or 6 or a value in between? 

10.  Line 154 – whilst 10 IBS-C patients were recruited in this study, it is worthwhile mentioning the potential limitations of the benefit of a low FODMAP diet in IBS-C which has been previously stated in the literature

11.  What was the primary endpoint for this study – this maybe helpful to clarify 

12.  Line 161 – There is a lot of mention of statistically significant differences in findings being stated in the manuscript – however, there is little mention of the clinical significance. For example, was the decrease in VAS bloating clinically significant? Also, Table 4 and 5 for example, state p-values, and no clinical values, making it difficult for potential readers to draw any conclusions 

13.  Line 202 – I presume this is IBS-D – please alter if so 

14.  Line 208 – it is worthwhile mentioning that healthy individuals have the same physiological responses to FODMAPs as individuals with IBS. Visceral hypersensitivity is the likely pathophysiological mechanism

15.  Line 225 – The authors state that a GFD may present a financial burden. It is worthwhile stating that GFD can be obtained from naturally occurring GF grains. There is also the statement of nutritional inadequacies of a GFD. It is worthwhile mentioning that most dietary therapies have the risk of nutritional inadequacies, and this is not unique to the GFD alone. In terms of meeting dietary requirements for example, it is known that the vast majority of individuals on a normal diet even fail to meet fibre recommendations.

16.  It is mentioned that the Mediterranean diet does not change the gut microbiota – please provide evidence to support this statement 

17.  Line 249 – The authors state that a long term low FODMAP diet cannot be recommended – however, this study has only looked at the short-term outcomes (4 weeks) of the low FODMAP diet and therefore I am unsure if this statement can be qualified from the data presented.

18.  In the conclusion, the authors suggest that a ‘balanced’ diet results in the spread of FODMAP content throughout the day – is there data from their study to support this, in terms of nutritional content analysis?

19.  Line 272 The authors state that the Mediterranean diet is easier to follow – it is worthwhile mentioning that from the data presented, this could be stated for short term but cannot be commented on in the long term. For example, it is unclear how the Mediterranean compares to the adapted low FODMAP diet in the long term 

Author Response

Dear reviewer,

Thank you very much for the constructive feedback you provided regarding our manuscript, “Effect of three diets (Low in FODMAPs, Gluten-free and Balanced) on Irritable Bowel Syndrome symptoms and health-related quality of life.”

As suggested the following articlehas undergone English language editing by MDPI. The text has been checked for correct use of grammar and common technical terms, and edited to a level suitable for reporting research in a scholarly journal. We agree with your suggestions and we tried to clarify as much as possible all the important issues you indicated:

1.     Line 49: we added reference entry n.7

2.     Line 52: we removed the statement

3.     Line 58: this is an interesting point and we added a comment on the study limitations incorporating part of your suggestion

4.     Line 60: the statement was not clear so we decided to remove it

5.     Line 70: that study was about a 6 weeks’ GFD diet and we rectified the statement

6.     Line 81: standard deviation has been added

7.     Line 85: we analysed dietary habits through 24 hrs recall

8.     Line 92: the diets were delivered by a dietitian and the low FODMAPs diet was in the strict reduction of all FODMAPs phase, we clarified this point in the manuscript

9.     Line 150 (now line 157): the correct basal mean value was 6

10.   Line 154 (now line 165): we mentioned the limitations incorporating part of you suggestion and added reference entry n.30

11.  Lines 74-77: we tried to clarify our aims

12.  We added clinical values in the tables 4 and 5

13.  Line 202 (now line 209): it was IBS D

14.  Line 208 (now line 221): we have incorporated your comment

15.  Line 225 (now line 239): This is an important issue and we tried to clarify this point. However, we still believe that GFD could in some cases be a financial burden because often patients, maybe because of scarce information or short free time, tend to buy gluten free products instead of trying to prepare themselves their meals with gluten free grains.

16.  Line 244 (now line 255): we highlighted some evidence about our statement with a reference entry

17.  Line 249: we removed the statement as ours is a short term study

18.  Conclusions: you have raised an important point. Although we can’t provide an accurate analysis about FODMAPs content per meal we hypothesize that Mediterranean diet, by avoiding an excessive calories charge per meal and by ensuring a balanced distribution of all nutrients throughout the day, prevents from an excessive charge of FODMAPs per meal. We tried to clarify this point

19.  Line 285: we specified that we can refer to short time intervals.

We hereby resubmit our manuscript for a secondary evaluation. Thank you once again for your consideration of our paper.

Sincerely

Dr. Danilo Paduano

Reviewer 3 Report

 The article is clear and well written. The research work is properly developed and it is interesting for treatment of IBS. Results are enough defined and discussion is adequate.

Nevertheless, some comments should been solved:

1.-It is not clear where the data of table 1 came from. It is not specified total energy intake, so it is not proper to present % of energy caming from each macronutrient without the data of daily calories burned +/- deviation.

2.- The authors don’t explain how and which kind of restriction have been made in the proposed “low FODMAP” model type diet. That is, they should explain which FODMAP restriction is made (lactose, fructose, fructans,… all of them?). In Gluten free diet and in the Balanced diet is easily conceivable or thinkable but not in the low FODMAP diet. They gave some advice to the patients, the software offered the low FODMAP diet (based on what criteria)

As a minor comment:

1- table 1 doesn’t reflect nutritional status, only the dietary habits or diet evaluation.

Author Response

Dear reviewer,

Thank you very much for the constructive feedback you provided regarding our manuscript, “Effect of three diets (Low in FODMAPs, Gluten-free and Balanced) on Irritable Bowel Syndrome symptoms and health-related quality of life.”

As suggested the following articlehas undergone English language editing by MDPI. The text has been checked for correct use of grammar and common technical terms, and edited to a level suitable for reporting research in a scholarly journal. We agree with your suggestions and we tried to clarify as much as possible all the important issues you indicated:

1. we specified the origin of the data in row 113, as you suggested we specified the average basal metabolic rate and the average daily calorie consumption in line 116, table 1 was completed with these data

2. the diets were delivered by a dietician and the low FODMAPs diet was in the strict reduction of all FODMAPs phase, we clarified this point in the manuscript in lines 92-98

3. we have modified the title of the table 1 as suggested

We hereby resubmit our manuscript for a secondary evaluation. Thank you once again for your consideration of our paper.

Sincerely

Dr. Danilo Paduano

Round 2

Reviewer 1 Report

I appreciate the clarification regarding ethics approval; the other issues, however, were not (probably could not be) adequately resolved.

Author Response

Dear Reviewer,

Thank you for your consideration and for the constructive feedback. We hereby tried to clarify the point about sample size, as to the other issues we made some changes in the manuscript as previously indicated, we’re afraid we didn’t reach out to adequately resolve them.

Sincerely,

Danilo Paduano

Reviewer 2 Report

The authors have taken on the comments mentioned on the manuscript and made extensive changes as suggested. An interesting manuscript, albeit with some limitations in methodology (all diets in same order without randomisation, although there was a washout period). This pragmatic real-life data adds to the growing evidence of dietary therapies in IBS. 

Author Response

Dear Reviewer,

Thank you for your consideration and for the constructive feedback. We hereby resubmit our manuscript.

Sincerely,

Danilo Paduano
